# Speaker-normalized sound representations in the human auditory cortex

Matthias J. Sjerps[1,2], Neal P. Fox [iD] [3], Keith Johnson[4] & Edward F. Chang[3,5]

The acoustic dimensions that distinguish speech sounds (like the vowel differences in "boot" and "boat") also differentiate speakers' voices. Therefore, listeners must normalize across speakers without losing linguistic information. Past behavioral work suggests an important role for auditory contrast enhancement in normalization: preceding context affects listeners' perception of subsequent speech sounds. Here, using intracranial electrocorticography in humans, we investigate whether and how such context effects arise in auditory cortex. Participants identified speech sounds that were preceded by phrases from two different speakers whose voices differed along the same acoustic dimension as target words (the lowest resonance of the vocal tract). In every participant, target vowels evoke a speaker-dependent neural response that is consistent with the listener's perception, and which follows from a contrast enhancement model. Auditory cortex processing thus displays a critical feature of normalization, allowing listeners to extract meaningful content from the voices of diverse speakers.

[1] Donders Institute for Brain, Cognition and Behaviour, Centre for Cognitive Neuroimaging, Radboud University, Kapittelweg 29, Nijmegen 6525 EN, The Netherlands. [2] Max Planck Institute for Psycholinguistics, Wundtlaan 1, Nijmegen 6525 XD, Netherlands. [3] Department of Neurological Surgery, University of California, San Francisco, 675 Nelson Rising Lane, San Francisco, California 94158, USA. [4] Department of Linguistics, University of California, Berkeley, 1203 Dwinelle Hall #2650, Berkeley, California 94720, USA. [5] Weill Institute for Neurosciences, University of California, San Francisco, 675 Nelson Rising Lane, San Francisco, California 94158, USA. Correspondence and requests for materials should be addressed to E.F.C. (email: edward.chang@ucsf.edu)

A fundamental computational challenge faced by perceptual systems is the lack of a one-to-one mapping between highly variable sensory signals and the discrete, behaviorally relevant features they reflect[1,2]. A profound example of this problem exists in human speech perception, where the main cues to speech sound identity also vary depending on speaker identity[3–5].

For example, to distinguish one speaker's /u/ ("boot") and /o/ ("boat"), listeners rely primarily on the vowel's first formant frequency (F1; the first vocal tract resonance, reflected as a dominant peak in the frequency spectrum) because it is lower for /u/ than for /o/[6]. However, people with long vocal tracts (typically tall male speakers) have overall lower resonance frequencies than those of speakers with shorter vocal tracts. Consequently, a tall person's production of the word "boat" and a short person's "boot" could be acoustically identical. Behavioral research has shown that preceding context allows listeners to tune-in to the acoustic properties of a particular voice and normalize subsequent speech input[7–11]. One classic example of this effect is that a single acoustic token, ambiguous between /u/ and /o/, will be labelled as /o/ after a context sentence spoken by a tall-sounding person (low F1), but as /u/ after a context sentence spoken by a shorter-sounding person (high F1)[9]. Therefore, understanding speech involves a process that builds up a representation of the characteristics of a speaker's voice and adjusts perception of new speech input to accommodate those characteristics.

There is considerable evidence that perceptually relevant sound representations arise within human parabelt (nonprimary) auditory cortex (AC). First, neural activity in the superior temporal gyrus (STG) is sensitive to acoustic-phonetic features, like F1, that are critical for recognizing and discriminating phonemes[12–18]. For example, vowels with low F1 frequencies (e.g., /u/, /i/) can be distinguished from vowels with relatively higher F1 frequencies (e.g., /o/, /æ/) based on local activity within human STG[19]. Second, the STG's encoding of speech is not a strictly linear (veridical) encoding the acoustics; rather, it reflects some properties of abstraction, including categorical perception, relative encoding of pitch, and attentional enhancement[13,20,21]. However, to date, it remains unknown whether speech representations in human AC also exhibit the type of context-dependence that could underlie speaker normalization.

Behavioral research in humans has previously suggested that normalization effects could partly arise from the general auditory contrast enhancement mechanisms[10,11,22–26], which are known to affect neurophysiological responses throughout the auditory hierarchy in animals[27–29]. Contrast enhancement models posit that adaptation to the frequency content of immediately preceding contexts—or their long-term average spectrum (see, e.g.,[11,30])—affects the responses to novel stimuli depending on the amount of overlap in their frequency content with that of the context. Moreover, behavioral evidence suggests that this contrast enhancement (and, hence, normalization) should arise—at least in part—centrally. For instance, contralateral (dichotic) presentation of the context (either speech or nonspeech) and target drive similar contrast enhancement effects in speech categorization as do ipsilaterally presented context and target[11,25,31,32].

Taken together, past work suggests a role for contrast enhancement in speech sound normalization in AC, but this prediction has not been directly demonstrated in neurophysiological studies in humans. Models of contrast enhancement make specific predictions about the responses of feature-tuned neuronal populations in AC[30]. Not only should the cortical representation of the same speech target depend on context, but, more specifically, context-dependent representations should differ in a particular (contrastive) way. That is, after a low-F1 context, the encoding of an ambiguous vowel target's F1 should more closely resemble the encoding of high F1 targets, while after a high F1 context it should more closely resemble low F1 targets. So far, contrast enhancement has been observed within frequency-tuned neurons in tonotopic primary AC in nonhuman mammals[27,28], but related patterns have not yet been observed in human AC, let alone in the context of human speech perception.

To investigate the influence of speaker context on speech sound encoding in AC, we recorded neural activity from human participants implanted with subdural high-density electrode arrays that covered peri-sylvian language cortex while they listened to and identified target vowels presented in the context of sentences spoken by two different voices[13,33]. We found direct evidence of speaker-normalized neural representations of vowel sounds in parabelt AC, including STG. Critically, the observed normalization effects reflected the contrastive relation between the F1 range in the context sentences and F1 of the target vowels, providing direct evidence for context-dependent contrast enhancement in human speech perception. More generally, the results demonstrate the critical role of human auditory-speech cortex in compensating for variability by integrating incoming sounds with their surrounding acoustic contexts.

## Results

**Speech sound perception is dependent on context**. We recorded neural activity directly from the cortical surface of five Spanish-speaking neurosurgical patients while they voluntarily participated in a speech sound identification task. They listened to Spanish sentences that ended in a (pseudoword) target, which they categorized as either "sufu" or "sofo" on each trial with a button press (Fig. 1a, b). The sentence-final targets comprised a digitally synthesized six-step continuum morphing from an unambiguous *sufu* to an unambiguous *sofo*, with four intermediate tokens (*s?f?*, i.e., spanning a perceptually ambiguous range). On each trial, a pseudo-randomly selected target was preceded by a context sentence (*A veces se halla…*; "At times she feels rather…"). Two versions of this sentence were synthesized, differing only in their mean F1 frequencies (Fig. 1a, c; Supplementary Fig. S1), yielding two contexts that listeners perceived as two different speakers: one with a long vocal tract (low F1; Speaker A) and one with a short vocal tract (high F1; Speaker B). Critically, F1 frequency is also the primary acoustic dimension that distinguishes between the vowels /u/ and /o/ in natural speech (in both Spanish and English) (Fig. 1a and Supplementary Fig. S1)[6]. Similar materials have previously been shown to induce a reliable shift in the perception of an /u/–/o/ continuum (a normalization effect) in healthy Spanish-, English-, and Dutch-listeners[7].

As expected, participants' perception of the target continuum was affected by the F1 range of the preceding sentence context ($\beta_{Context\ F1} = -1.70$, $t = -3.51$, $p < 0.001$; Fig. 1d; see Supplementary Materials for details of the mixed effects logistic regression that was used). Specifically, participants were more likely to identify tokens as *sofo* (the vowel category corresponding to higher F1 values) after a low-F1 voice (Speaker A) compared to the same target presented after a high-F1 voice (Speaker B). Hence, listeners' perceptual boundary between the /u/ and /o/ vowel categories shifted to more closely reflect the F1 range of the context speaker. Past work has interpreted this classical finding in light of the contrastive perceptual effects that are ubiquitous among sensory systems[34]: the F1 of a speech target will sound relatively higher (i.e., sound more like an /o/) after a low-F1 context sentence than after a high-F1 context. Behaviorally, this is reflected as a shift of the category boundary to lower-F1 values.

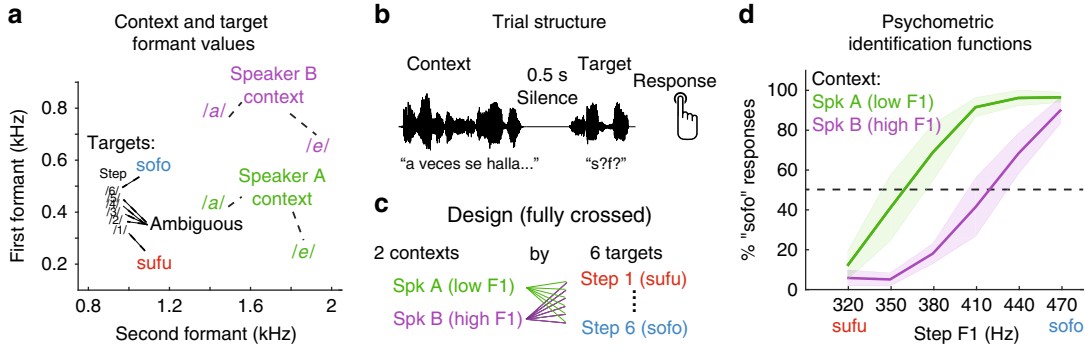

**Fig. 1** Listeners perceive speech sounds relative to their acoustic context. **a** Target sounds were synthesized to create a six-step continuum ranging from *sufu* (step 1; low-first formant [F1]) to *sofo* (step 6; high F1). Context sentences were synthesized to sound like two different speakers: a speaker with a long vocal tract (low-F1 range: Speaker A), and a speaker with a short vocal tract (high-F1 range; Speaker B). Context sentences contained only the vowels /e/ and /a/, but not the target vowels /u/ and /o/. **b** Context sentences preceded the target on each trial (separated by 0.5 s of silence), after which participants responded with a button press to indicate whether they heard "sufu" or "sofo". **c** All targets were presented after both speaker contexts. **d** Listeners more often gave "sofo" responses to target sounds if the preceding context was spoken by Speaker A (low F1) than Speaker B (high F1). Error bars indicate s.e.m.

**Human AC exhibits speaker-dependent speech representations**. Two of the most influential hypotheses explaining the phenomenon of speaker normalization posit that: (1) contrast enhancing processes, operating at early auditory processing levels, change the representation of the input signal in a normalizing way[11,23,30,31]; (2) alternatively, it has been suggested that normalization does not alter the perceptual representation of speech, but, instead, that normalization arises as a consequence of a speaker-specific mapping of the auditory representation onto abstract linguistic units (i.e., listeners have learned to map an F1 of 400 Hz to /u/-words for speakers, or vocal tracts, that sound short, but to /o/-words for speakers that sound taller)[35,36]. Hence, while the contrast enhancement hypothesis predicts normalized representations in AC, the latter theory predicts that early auditory representations of speech cues remain unnormalized (i.e., independent of speaker-context).

Past neurobiological work has demonstrated that neural populations in parabelt AC are sensitive to acoustic-phonetic cues that distinguish classes of speech sounds, including vowels, and not to specific phonemes per se[19]. Hence, the primary goal of the current study was to examine whether the neural representation of vowels in parabelt AC shifts in a contrast enhancing way relative to the acoustic characteristics of the preceding speaker, or, alternatively, whether such contrast enhancement (or normalization) is not reflected in parabelt AC processing. We first tested whether individual cortical sites that reliably differentiate between vowels (i.e., discriminate /u/ from /o/ in their neural response) exhibit normalization effects.

To this end, we examined stimulus-locked neural activity in the high-gamma band (70–150 Hz) at each temporal lobe electrode ($n = 406$ across patients; this number is used for all Bonferroni corrections below) during each trial. High-gamma activity is a spatially and temporally resolved neural signal that has been shown to reliably encode phonetic properties of speech sounds[19,37,38], and is correlated with local neuronal spiking[39–41]. We used general linear regression models to identify local neural populations involved in the representation of context and/or target acoustics. Specifically, we examined the extent to which high-gamma activity at each electrode encoded stimulus properties during presentation of the context sentences (context window) or during presentation of the target (target window; see Supplemental Materials). The fully specified encoding models included numerical predictors for the target vowel F1 (steps 1–6) and context F1 (high vs. low), as well as their interaction. In the following, we focused on task-related electrodes, defined as the subset of temporal lobe electrodes for which a significant portion of the variance was explained by the full model, either during the target window or during the context window ($p < 0.05$; uncorrected, $n = 98$; see Supplementary Fig. S2).

Among the task-related electrodes, a subset displayed selectivity for target vowel F1 (Fig. 2a: electrodes displaying a main effect for target F1). Consistent with previous reports of AC tuning for vowels[19], we observed that different subsets of electrodes displayed a preference for either *sufu* or *sofo* targets (color-coded in Fig. 2a). Fig. 2b and Fig. 2c (middle panel) display the response profile for one example electrode that had a *sofo* preference (e1; $\beta_{\text{Target F1}} = 2.80$, $t = 9.34$, $p = 1.10 \times 10^{-18}$). Importantly, in addition to overall tuning to the target sound F1, the activation level of this electrode was modulated by the F1 range of the *preceding* context (Fig. 2b and bottom panel of Fig. 2c; $\beta_{\text{Context F1}} = -2.23$, $t = -4.51$, $p = 8.3 \times 10^{-6}$). This demonstrates that the responsiveness of a neural population that is sensitive to bottom-up acoustic cues is also affected by the distribution of that cue in a preceding context. The direction of this influence is the same as the behavioral normalization effect, such that a low-F1 speaker context was associated with stronger responses for *sofo* (high-F1) targets.

To quantify this normalization effect across all electrodes that display selectivity to target acoustics, we used linear mixed effects regression to estimate the relation between electrodes' target preference (defined as the glm-based signed $t$-statistic of the target F1 factor during the target window) and their context effect (defined as the glm-based signed $t$-statistic of the context F1 factor during the target window) (see Supplementary Materials for further detail on this analysis). We found that the magnitude and direction of an electrode's context effect was predicted by the magnitude and direction of its target preference (Fig. 2d). Crucially, this strong relationship had a negative slope, such that electrodes that had high-F1 target preferences (*sofo* > *sufu*) had stronger responses to targets after low-F1 context sentences (low-F1 context > high-F1 context; $\beta_{\text{Targ-F1-t}} = -0.32$, $t = -5.00$, $p = 1.53 \times 10^{-5}$). Importantly, this demonstrates that the relationship between context response and target response reflects an encoding of the contrast between the formant properties of each, consistent with the normalization pattern observed in the behavioral responses (Fig. 1d) and with the predictions of a contrast enhancement model of speech normalization.

**Normalization of vowel representations in all participants**. Figure 2d demonstrates that local populations in AC that display tuning for specific target vowel F1 ranges exhibit normalization.

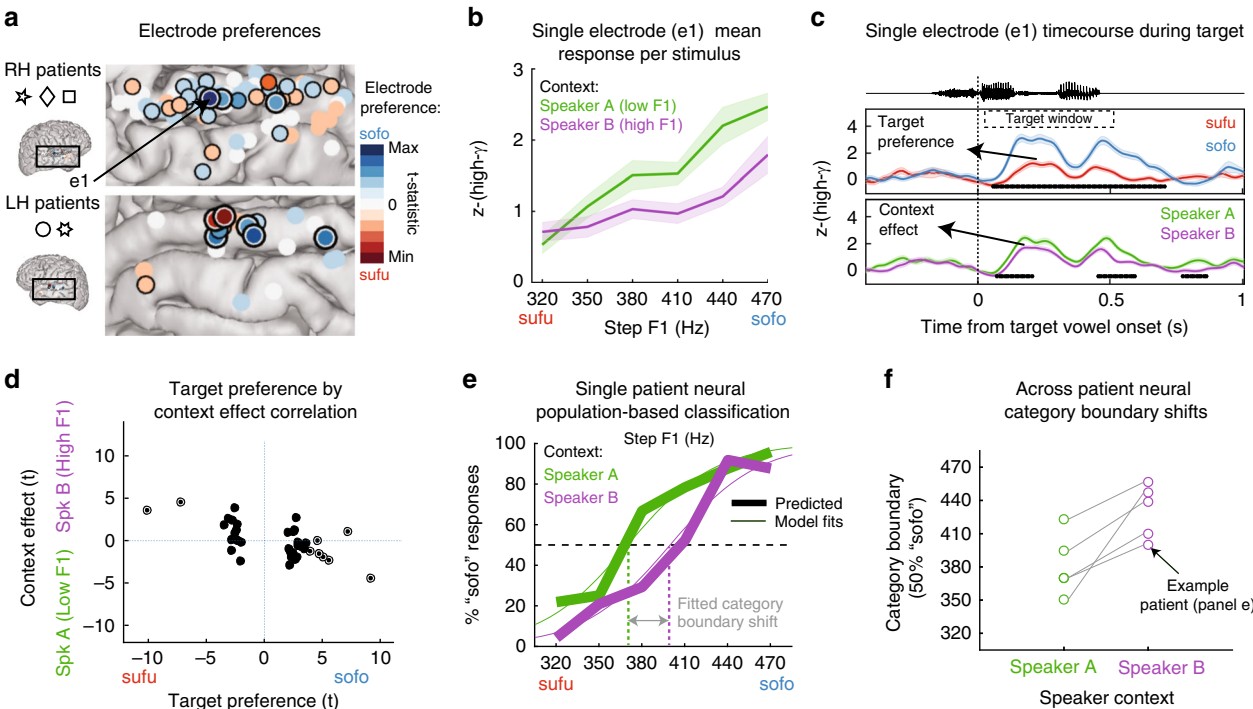

**Fig. 2** The neural response to bottom-up acoustic input is modulated by preceding context. **a** Target vowel preferences and locations (plotted on a standardized brain) for electrodes from all patients (three with right hemisphere [RH] and two with left hemisphere [LH] grid implants). Only those temporal lobe electrodes where the full omnibus model was significant during the context and/or the target window (F-test; $p < 0.05$) are displayed. Strong target F1 selectivity is relatively uncommon: electrodes with a black-and-white outline are significant at Bonferroni corrected $p < 0.05$ ($n = 9$, out of 406 temporal lobe electrodes); a single black outline indicates significance at a more liberal threshold ($p < 0.05$, uncorrected; $n = 28$). Activity from the indicated electrode (e1) is shown in **b** and **c**. **b** Example of normalization in a single electrode (e1; z-scored high-gamma [high-γ] response averaged across the target window [target window marked in **c**]). **c** Activity from e1 across time, separating the endpoint targets (middle panel) or the contexts (bottom panel). The electrode responds more strongly to /o/ stimuli than /u/ stimuli, but also responds more strongly overall after Speaker A (low-F1). This effect is analogous to the behavioral normalization (Fig. 1d). Black bars at the bottom of the panels indicate significant time-clusters (cluster-based permutation test of significance). **d** Among all electrodes with significant target sound selectivity ($n = 37$ [9 + 28]), a relation exists between the by-electrode context effect and target preference. Both are expressed as a signed t-value, demonstrating that the size and direction of the target preferences predicts the size and direction of the context effects. **e** An LDA classifier was trained on the distributed neural responses elicited by the *sufu* and *sofo* endpoint stimuli using all task-related electrodes. This model was then used to predict classes for neural responses to (held-out) endpoint tokens and for the ambiguous steps in each context condition. Proportions of neurally based "sofo" predicted trials (thick lines) display a relative shift between the two context conditions (data from one example patient). Regression curves were fitted to these data for each participant separately to estimate 50% category boundaries per condition for panel **f** (thin lines). **f** The neural classification functions display a shift in category boundaries between context conditions for all patients individually. In **b** and **c**, error bars indicate ±1 s.e.m.

However, only a few electrodes ($n = 9$, out of 406 temporal lobe electrodes) displayed very strong tuning (significance at Bonferroni-corrected $p < 0.05$), while the majority of F1-tuned electrodes displayed only moderate tuning and moderate context effects. Moreover, not all participants had electrodes that displayed strong target F1 tuning (see Table S2). The relative sparseness of strong tuning is not surprising given that the target vowel synthesis involved only small F1 frequency differences (~30 Hz per step), with the endpoints being separated by only 150 Hz (which is, however, a prototypical F1 distance between /u/ and /o/[7]). However, past work has demonstrated that even small acoustic differences among speech sounds are robustly encoded by distributed patterns of neural activity across AC[13,14]. In order to determine whether distributed neural representations of vowels reliably display normalization across all participants, we trained a multivariate pattern classifier model (linear discriminant analysis, LDA) on the spatiotemporal neural response patterns of each participant. Models were trained to discriminate between the endpoint stimuli (i.e., trained on the neural responses to steps 1 vs. 6, irrespective of context) using all task-related electrodes for that participant. These models were then used to predict labels for held-out neural responses to both the endpoints and the ambiguous steps in each context condition. For all participants, classification of held-out endpoint trials was significantly better than chance (Supplementary Fig. S3b). To assess the influence of target F1 and context F1 on the classifier output, a logistic generalized linear mixed model was then fit to the proportion of predicted *sofo* responses across all participants.

Figure 2e displays the proportion of *sofo* labels predicted for all stimuli by the LDA classifier based on the neural data of one example participant (thick lines). Importantly, a shift is observed in the point of crossing of the category boundary. Regression functions fitted to these data (thin lines) were used to estimate the size and direction of the context-driven neural boundary (50% crossover point) shift per participant. For each participant, the neural vowel boundaries, like the behavioral vowel boundaries, were found to be context-dependent (Fig. 2f; see Supplementary Materials and Supplementary Fig. S3 for further detail).

A combined regression analysis demonstrated that, across participants, population neural activity in the temporal lobe was modulated both by the acoustic properties of the target vowel ($\beta_{Target\ F1} = 0.50$, $t = 13.20$, $p < 0.001$) and by the preceding

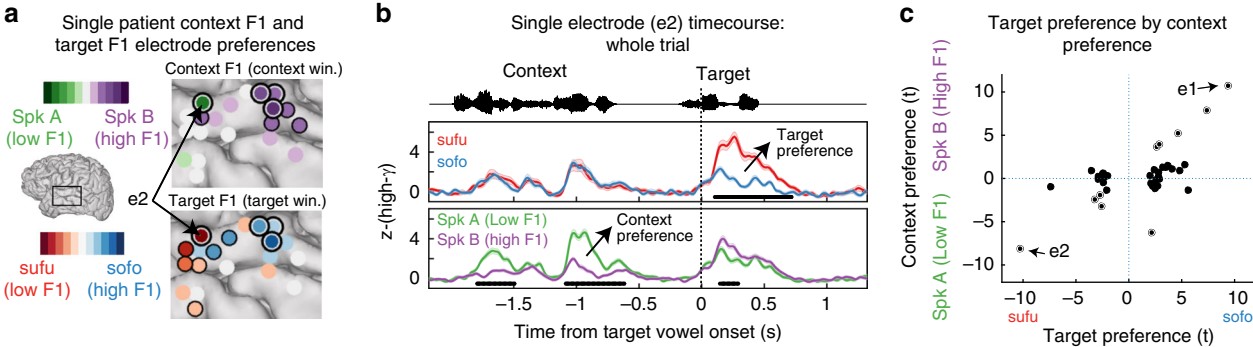

**Fig. 3** Sensitivity to contrast in acoustic-phonetic features. **a** Electrode preferences for both context F1 (during context window) and target F1 (during target window) from a single example patient. Some populations display both target F1 selectivity and context F1 selectivity (marked with a black-and-white outline), indicating a general preference for higher or lower F1 frequency ranges. Others are only tuned for target F1 or context F1 (marked with a single black outline in their respective panels). Significance assessed at $p < 0.05$ uncorrected. **b** Mean (±1 s.e.m.) high-gamma activity at an example electrode (e2) from the example patient in panel **a** (conditions split as described in Fig. 2c). Activity is displayed for a time window encompassing the full trial duration (both precursor sentence and target word). Black bars represent significant time-clusters ($p < 0.05$; cluster-based permutation). **c** A relation exists between the by-electrode context preference and target preference: electrodes that display a preference for either high- or low-target F1 typically also display a preference for the same F1 range during the context

context ($\beta_{\text{Context F1}} = -0.34$, $t = -5.58$, $p < 0.001$). Moreover, this effect was not observed for task-related electrodes outside of the temporal lobe during the target window (see Supplementary Fig. S4; nontemporal electrodes were mostly located on sensorimotor cortex and the inferior frontal gyrus).

Importantly, and consistent with participants' perception, the neural classification functions demonstrate that the influence of the context sentences consistently affected target vowel representations in a contrastive (normalizing) direction: the neural response to an ambiguous target vowel with a given F1 is more like that of /o/ (high-F1) after a low-F1 context (Speaker A) than after a high-F1 context (Speaker B; see Supplementary Figs. S3 and S4b for more detail).

**Normalization by acoustic-phonetic contrast enhancement**. It has been suggested that a major organizing principle of speech encoding by human parabelt AC is its encoding of acoustic-phonetic features, which are more cross-linguistically generalizable and more physically grounded than phonemes (or other possible higher-level linguistic representations) per se[12,19,42–44]. However, AC processing is diverse and may contain regions that are, in fact, selective for (more abstract) phonemes. For example, AC has also been found to display properties that are typically associated with abstract sound categories such as categorical perception[13]. Hence, we next assessed whether the normalization effects observed here involved neural populations that display sensitivity to acoustic-phonetic features (i.e., relating to more general F1 characteristics). Because the context sentence ("*A veces se halla…*") did not contain the target vowels /u/ or /o/, while its F1 values did cover the same general frequency range, we examined neural responses during the context window to understand individual electrode preferences.

To this end we again used the glm-based *t*-statistics of all temporal lobe electrodes that displayed tuning for the endpoint vowels ($n = 37$; as per Fig. 2d). Among these electrodes, however, we examined the relationship between their preferences for context F1 during the context window and for target F1 during the target window. Figure 3a displays context and target preferences on the cortex of a single-example patient. Among the electrodes that displayed target F1 selectivity, some also displayed selectivity for the context F1 during the context window (such dual preferences are indicated with a black-and-white outline). Figure 3b displays the activation profile of one example

electrode (e2). Importantly, e2 responded more strongly to low-F1 targets during the target window (*sufu* preference: $\beta_{\text{Target F1}} = -0.52$, $t = -10.25$, $p = 3.9 \times 10^{-21}$), but also to low-F1 contexts during the context window (Speaker A preference: $\beta_{\text{Target F1}} = -0.54$, $t = -8.05$, $p = 2.3 \times 10^{-14}$). This demonstrates that this neural population responded more strongly to low-F1 acoustic stimuli in general and is not exclusively selective for a discrete phoneme category. Importantly, e2 also displayed normalization, as its activity was affected by context F1 during the target window ($p = 2.7 \times 10^{-4}$), and the direction of that context effect was consistent with contrastive normalization (cf. Fig. 2d).

Extending this finding to the population of electrodes, we found a significant positive relation across all target-tuned temporal lobe electrodes between an electrode's target preference and its context preference ($\beta_{\text{Cont-F1}-t} = 0.76$, $t = 3.59$, $p = 9.83 \times 10^{-4}$; Fig. 3c; see Supplementary Materials for more detail on this analysis). Hence, a neural population's tuning for higher or lower F1 ranges tended to be general, not vowel-specific. Moreover, when restricting the test of normalization (assessed as the relationship between target preferences and the context effect, as per Fig. 2d) to those electrodes that displayed significant tuning for both target F1 and context F1, robust normalization was again found (Supplementary Fig. S5). These findings confirm that normalization affects acoustic-phonetic (i.e., pre-phonemic) representations of speech sounds in parabelt AC.

## Discussion

A critical challenge that listeners must overcome in order to understand speech is the fact that different speakers produce the same speech sounds differently, a phenomenon that is known as the lack-of-invariance problem in speech perception[1,3]. This issue is partly due to the fact that different speakers' voices span different formant ranges. We investigated the neural underpinnings of how listeners use speaker-specific information in context to normalize phonetic processing. First, we observed behavioral normalization effects, replicating previous findings[7–9,24]. More importantly, we observed normalized representations of vowels in parabelt AC. These normalized representations were observed broadly across parabelt AC and were observed for all participants individually. Moreover, we found that these effects follow the predictions of a general auditory contrast enhancement model of normalization[30], affecting speech sound representations at a level that precedes the mapping onto phonemes or higher-level

linguistic units. These findings suggest that contrast enhancement plays an important role in normalization.

Recent research has demonstrated that AC responds to acoustic cues that are critical for both recognizing and discriminating phonemes[12,13,15–18,45] and different speakers[46–50]. However, since cues that are critical for speaker and phoneme identification are conflated in the acoustic signal, such findings could be consistent with either context-dependent or context-independent cortical representations of acoustic properties. Our experiments demonstrate that rapid and broadly distributed normalization through contrast enhancement is a basic principle of how human AC encodes speech.

Qualitatively similar contrast enhancing operations have been widely documented in animal neurophysiological research, where it has been demonstrated to involve neural mechanisms such as adaptive gain control[27–29,51] or stimulus specific adaptation[28,29]. An intuitive mechanism for the implementation of contrast enhancement that follows from that work involves sensory adaptation. This could be based on neuronal fatigue: when a neuron, or neuronal population, responds strongly to a masker stimulus, its response to a subsequent probe is often attenuated when the frequency of the probe falls within the neurons' excitatory receptive field[52,53]. But in addition to such local forms of adaptation, adaptation has also been thought to arise through (inhibitory) interactions between separate populations of neurons (which may have partly non-overlapping receptive fields)[27,51]. In the present study, spectral peaks in the two context sentences and those in the endpoint target vowels were partly overlapping (see Supplementary Fig. S1). These forms of adaptation may, hence, play a role in the type of normalization observed here. Indeed, we observed a number of populations for which a strong preference for one of the context sentences during the context window was associated with a decreased response during the target window (i.e., the normalization effect; Fig. 3b).

The prediction that contrast enhancement may play an important role in human speech sound normalization was previously made based on behavioral studies on contrastive context effects in speech perception[10,11,23,30,34,54]. A relevant observation from that literature is that, under specific conditions, nonspeech context sounds (e.g., broadband noise and musical tones) have also been observed to affect the perception of speech sounds[11,22,23,31,55]. This interpretation has been challenged, however, suggesting that speech- and nonspeech-based context effects could be based on qualitatively different processes[56,57]. We did not test the influence of nonspeech contexts here, and such an investigation would provide an important next step in the study of context effects in speech perception. However, our findings most closely align with a model that assumes that normalization effects may not be speech-specific and that normalization can, at least in part, be explained by more general auditory adaptation effects[23,34,58].

An interesting additional question concerns the main locus of emergence of normalization. Broadly speaking, normalization could be inherited from primary AC or subcortical regions (from which we were unable to record; see[59] for a more detailed discussion of these potential influences); it may largely emerge within parabelt AC itself; or it could be driven by top-down influences from regions outside of the AC. In our study, context and target sounds were separated in time by a 500 ms silent interval. It has been suggested that, over such relatively long latencies, adaptation effects become especially dominant at cortical levels of processing but are reduced at more peripheral levels of processing[55,60,61]. Furthermore, behavioral experiments have demonstrated robust normalization effects with contralateral presentation of context and target sounds[11,31,32] (i.e., a procedure that reduces precortical interaural interactions). Both observations thus suggest that

normalization can also arise when the contribution of context-target interactions in the auditory periphery may be limited. With respect to the potential role of top-down modulations from regions outside of the AC, inferior frontal and sensorimotor cortex have been suggested to be involved in the resolution of perceptual ambiguities in speech perception[62,63] and could, hence, have been expected to play a role in normalization, too. Here, we observed considerable activation in these regions, but, intriguingly, they did not display normalization during the processing of the target sounds (see Supplementary Fig. S4). While tentative, these combined findings highlight human AC as the most likely locus for the emergence of the context effects in speech processing observed here.

In the current experiment, we recorded neural activity from cortical sites in both the left and right hemispheres. It has previously been demonstrated that the right hemisphere is more strongly involved in the processing of voice information[64,65]. Here, normalization was observed in every participant, irrespective of which hemisphere was the source of a given participant's recordings (Fig. 2f). Importantly, however, recordings from any given participant only included measurements from a single hemisphere, so no strong conclusions regarding lateralization should be drawn based on this dataset.

Despite normalization of vowel representations, responses were not completely invariant to speaker differences during the context sentences (see, for example, the behavior of example electrode e2 in Fig. 3b, which displays a preference for the Low F1 sentence throughout most of the context window: i.e., it is not fully normalized). And indeed, our (and previous[7,11,36]) findings show that, even for target sound processing, surrounding context rarely (if ever) results in complete normalization. That is, while context sentence F1 differed by roughly 400 Hz between the two speakers, the normalization effect only induced a shift of ~50 Hz in the position of the category boundaries (in behavior and in neural categorization). Moreover, our target F1 values were ideally situated between the two context F1 ranges, which raises the question of whether equally large effects would have been observed for other target F1 ranges. The role of contrast enhancement in normalization should thus be seen as a mechanism that biases processing in a context-dependent direction, but not one that fully normalizes processing. Furthermore, context-based normalization is not the only means by which listeners tune-in to specific speakers: listeners categorize sound continua differently when they are merely told they are listening to a man or a woman, demonstrating the existence of normalization mechanisms that do not rely on acoustic contexts (and hence acoustic contrast) at all[66]. In addition, formant frequencies are perceived in relation to other formants and pitch values in the current signal, because those features are correlated within speakers (e.g., people with long vocal tracts typically have lower pitch and lower formant frequencies, overall). These intrinsic normalization mechanisms have been shown to affect AC processing of vowels as well[67–71]. Tuning-in to speakers in everyday listening must thus result from a combination of multiple mechanisms, involving at least these three distinct types of normalization[8].

To conclude, the results presented here reveal that the processing of vowels in AC becomes rapidly influenced by speaker-related acoustic properties in preceding context. These findings add to a recent literature that is ascribing a range of complex acoustic integration processes to the broader AC, suggesting that it participates in high-level encoding of speech sounds and auditory objects[13,19,72–74]. Recently, it has been demonstrated that populations in parabelt AC encode speaker-invariant contours of intonation that speakers use to focus on one or the other part of a sentence[20]. The current findings build on these and

demonstrate the emergence of speaker-normalized representations of acoustic-phonetic features, the most fundamental building blocks of spoken language. This context-dependence allows AC to partly resolve the between-speaker variance present in speech signals. These features of AC processing underscore its critical role in our ability to understand speech in the complex and variable situations that we are exposed to every day.

## Methods

**Patients.** A total of five human participants (2 male; all right-handed; mean age: 30.6 years), all native Spanish-speaking (the US hospital at which participants were recruited has a considerable Spanish-speaking patient population), were chronically implanted with high-density (256 electrodes; 4 mm pitch) multi-electrode cortical surface arrays as part of their clinical evaluation for epilepsy surgery. Arrays were implanted subdurally on the peri-Sylvian region of the lateral left ($n = 2$) or right ($n = 3$) hemispheres. Placement was determined by clinical indications only. All participants gave their written informed consent before the surgery, and had self-reported normal hearing. The study protocol was approved by the UC San Francisco Committee on Human Research. Electrode positions for reconstruction figures were extracted from tomography scans and co-registered with the patient's MRI.

**Stimulus synthesis.** Details of the synthesis procedure for these stimuli have been reported previously[7]. All synthesis was implemented in Praat software[75]. In brief, using source-filter separation, the formant tracks of multiple recordings of clear "sufu" and "sofo" were estimated. These estimates were used to calculate a single average time-varying formant track for both words, now representing an average of the formant properties over a number of instances of both [o] and [u]. The height of only the first formant of this filter model was increased and decreased across the whole vowel to create the new formant models for the continuum from [u] to [o] covering the distance between endpoints in six steps. These formant tracks were combined with a model of the glottal-pulse source to synthesize the speech sound continuum. Synthesis parameters thus dictated that all steps were equal in pitch contour, amplitude contour and had identical contours for the formants higher than F1 (note that F1 and F2 values in Fig. 1a and S1 reflect measurements of the resulting sounds, not synthesis parameters). The two context conditions were created through source-filter separation of a single spoken utterance of the sentence "*a veces se halla*" ("*at times she feels rather…*"). The first formant of the filter model was then increased or decreased by 100 Hz and recombined with the source model following similar procedures as for the targets.

**Procedures.** The participants were asked to categorize the last words of a stimulus as either sufu or sofo. Listeners responded using the two buttons on a button box. The two options sufu and sofo were always displayed on the computer screen. Each of the 6 steps of the continuum was presented in both the low- and high-F1 sentence conditions. Context conditions were presented in separate mini-blocks of 24 trials (6 steps × 4 repetitions). Participants participated in as many blocks as they felt comfortable with (see Table S1 for trial counts).

**Data acquisition and preprocessing.** Cortical local field potentials were recorded and amplified with a multichannel amplifier optically connected to a digital signal acquisition system (Tucker-Davis Technologies) sampling at 3052 Hz. The stimuli were presented monaurally from loudspeakers at a comfortable level. The ambient audio (recorded with a microphone aimed at the participant) along with a direct audio signal of stimulus presentation were simultaneously recorded with the ECoG signals to allow for precise alignment and later inspection of the experimental situation. Line noise (60 Hz and harmonics at 120 and 180 Hz) was removed from the ECoG signals with notch filters. Each time series was visually inspected for excessive noise, and trials and or channels with excessive noise or epileptiform activity were removed from further analysis. The remaining time series were common-average referenced across rows of the 16 × 16 electrode grid. The time-varying analytic amplitude was extracted from eight bandpass filters (Gaussian, with logarithmically increasing center frequencies between 70 and 150 Hz, and semilogarithmically increasing bandwidths) with the Hilbert transform. High-gamma power was calculated by averaging the analytic amplitude across these eight bands. The signal was subsequently downsampled to 100 Hz. The signal was z-scored based on the mean and standard deviation of a baseline period (from −50 to 0 ms before the onset of the context sentence) on a trial by trial basis. In the main text, high-γ will refer to this measure.

**Single-electrode encoding analysis.** We used ordinary least-squares linear regression to predict neural activity (high-γ) from our stimulus conditions (target F1 steps, coded as −2.5, −1.5, −0.5, 0.5, 1.5, 2.5; and context F1, coded as −1 and 1; as well as their interaction). These factors were used as numerical predictors to neural activity that was averaged across the target window (from 70 to 570 ms after target vowel onset) or across the context window (from 250 to 1450 ms after context sentence onset—a later onset was chosen to reduce the influence of large

and non-selective onset responses present in some electrodes). For each model, R-squared ($R^2$) provides a measure of the proportion of variance in neural activity that is explained by the complete model. The p value associated with the omnibus F-statistic provides a measure of significance. We set the significance threshold at alpha = 0.05 and corrected for multiple comparisons using the Bonferroni method, taking individual electrodes as independent samples. Supplementary Figure S2a, b demonstrates that most of the variance in the context was explained by the factor context F1. During the target window, however, both target F1 and context F1 explain a considerable portion of the variance. The interaction term was included to accommodate a situation where the context effect is more strongly expressed on one side of the target continuum than the other (see e.g., Fig. 2b, where the context effect is larger toward sofo), but is not further interpreted here.

For Fig. 2d and Fig. 3c, linear mixed regression analyses were used to assess the relation between signed t-statistics of target F1 preferences and context effects (Fig. 2d) or context preferences (Fig. 3c). Regression estimates were computed over all significant (9 [corrected] + 28 [uncorrected] = 37) electrodes. Linear mixed effects regression accommodates the hierarchical nature of these observations (electrodes within patients).

**Cluster-based permutation analyses.** For single-example electrodes, a cluster-based permutations approach was used to assess significance of differences between two event related high-gamma time courses (Fig. 2c and Fig. 3b; following the method described in ref. [76]). For each permutation, labels of individual trials were randomly assigned to data (high-gama time courses), and a t test was performed for each timepoint. Next, for each time point (across all 1000 permutations) a criterion value was established (the highest 95% of the [absolute] t values for that timepoint). Then, for each permutation, it was established when its value reached above the criterion value and for how many samples it remained above criterion. A set of subsequent timepoints above criterion is defined as a cluster. Then, for each cluster the t values were summed, and this value was assigned to that entire cluster. For each permutation only the largest (i.e., highest summed cluster value) was stored as a single value. This resulted in a distribution of maximally 1000 cluster values (some permutations may not result in any significant cluster and have a summed t value of 0). Then, using the same procedure, the size of all potential clusters was established for the real data (correct assignment of labels), and it was established whether the size of each cluster was larger than 95% of the permutation-based cluster values. $p < 0.001$ indicates that the observed cluster was larger than all permutation based clusters.

**Stimulus classification.** Linear discriminant analysis (LDA) models were trained to predict the stimulus from the neural population responses evoked by the stimuli. Per participant a single model was trained on all endpoint data, which was then used to predict labels for the ambiguous items. To predict stimulus class for the endpoint stimuli (steps 1 and 6) a leave-one-out cross validation procedure was used to prevent overfitting. Model features (predictors) consisted of the selected timepoint*electrode combinations per participant.

For the analyses (Fig. 2; Supplementary Fig. S3; Supplementary Fig. S4) training data consisted of high-γ data averaged across a 500 ms time window starting 70 ms after target vowel onset (the target vowel was the first point of acoustic divergence between targets).

In the analyses, all task-related electrodes for a given participant (and region-of-interest, see Fig. S4) were selected. Trial numbers per participant are listed in Table S1. The analysis displayed in Fig. 2 and Supplementary Figs. S3 and S4 hence relied on a large number of predictors (electrodes × timepoints). While a large amount of predictors could result in overfitting, these parameters led to the highest proportion of correct classification for the endpoints: 76% correct (see Supplementary Fig. S1b), although note that this number may be inflated because of electrode pre-selection. This approach was chosen, however, because high endpoint classification performance is important to establish the presence of normalization: a shift in a response function can only be detected if the steepness of that function is nonzero. Importantly, specifically selecting electrodes that distinguish the endpoints does not affect the extent of observed normalization, because the normalization effect is orthogonal to that of target F1 (i.e., normalization itself was not a selection criterion). Furthermore, in all analyses, classification scores were only obtained from held-out data, preventing the fitting of idiosyncratic models. In addition, averaging across time (hence decreasing the number of predictors) led to qualitatively similar (and significant) effects for the important comparisons reported in this paper. Classification analyses resulted in a predicted class for each trial. These data were used as input for a generalized logistic linear mixed effects model.

**Linear mixed effects regression of classification data.** For the analyses that assessed the effects of target stimulus F1 and context F1 on proportion of "sofo" responses (both behavioral and neural-classifier-based) we employed generalized linear mixed effects models (glmer; with the dependent variable "family" set to "binomial"). The models had Target F1 (contrast coded, with the levels −2.5; −1.5; −0.5; 0.5; 1.5; 2.5) and Context F1 (levels −1; 1) entered as fixed effects, and uncorrelated by-patient slopes and intercepts for these factors as random effects.

For the analysis of the behavioral data ($\beta_{\text{Intercept}} = 0.52$, $t = 0.79$, $p = 0.42$), we observed more *sofo* responses towards the *sofo* end of the stimulus continuum ($\beta_{\text{Target F1}} = 1.86$, $t = 4.10$, $p < 0.001$). Moreover, we observed an effect of context as items along the continuum were more often perceived as *sofo* (the vowel category corresponding to higher-F1 values) after a low-F1 voice (Speaker A) than after a high-F1 voice (Speaker B), ($\beta_{\text{Context F1}} = -1.70$, $t = -3.51$, $p < 0.001$).

For the analyses of neural representations the dependent variable consisted of the classes predicted by LDA stimulus classification described above. For the overall analysis including temporal lobe electrodes, the model ($\beta_{\text{Intercept}} = -0.16$, $t = -1.06$, $p = 0.29$) revealed significant classification of the continuum ($\beta_{\text{Target F1}} = 0.50$, $t = 13.20$, $p < 0.001$), suggesting reliable neural differences between the endpoints. Furthermore, an effect was also found for the factor Context on the proportion of "sofo" classifications ($\beta_{\text{Context F1}} = -0.34$, $t = -5.58$, $p < 0.001$), reflecting the normalization effect of most interest. For the analysis focusing on the dorsal and frontal electrodes ($\beta_{\text{Intercept}} = -0.16$, $t = 0.91$, $p = 0.37$) a significant effect of Step was observed; that is, significant classification of the continuum ($\beta_{\text{Target F1}} = 0.23$, $t = 7.01$, $p < 0.001$), but no significant influence of context ($\beta_{\text{Context F1}} = -0.02$, $t = -0.39$, $p = 0.69$) see Supplementary Fig. S4C for further detail.

**Reporting summary**. Further information on research design is available in the Nature Research Reporting Summary linked to this article.

## Data availability

The data that support the findings of this study are publicly available through the Open Science Framework at https://osf.io/t87d2/.

## Code availability

These results were generated using code written in Matlab. Code is publicly available through the Open Science Framework at https://osf.io/t87d2/.

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

## Acknowledgements

We are grateful to Matthew K. Leonard for commenting on an earlier version of this manuscript and to all members of the Chang Lab for helpful comments throughout this work. This work was supported by European Commission grant FP7-623072 (M.J.S.); and NIH grants R01-DC012379 (E.F.C.) and F32-DC015966 (N.P.F.). E.F.C. is a New York Stem Cell Foundation-Robertson Investigator. This research was also supported by The William K. Bowes Foundation, the Howard Hughes Medical Institute, The New York Stem Cell Foundation and The Shurl and Kay Curci Foundation.

## Author contributions

M.J.S. and K.J. conceived the study. M.J.S. designed the experiments, generated the stimuli, and analyzed the data. M.J.S., N.P.F. and E.F.C. collected the data. M.J.S. and N.P.F. interpreted the data and wrote the paper. M.J.S., N.P.F., K.J. and E.F.C. edited the paper.

## Additional information

**Competing interests:** The authors declare no competing interests.

