## [Peer Review File · Nature Communications]

Reviewers' Comments:

Reviewer #1:

Remarks to the Author:

The major finding of the paper is that acoustically-identical speech sounds are encoded in a context-dependent manner in bilateral human temporal cortex. I am enthusiastic about the work and I find the study and its analysis to be well-conceptualized and carefully conducted. The data are sparse (few electrodes, few subjects), but the replication of the results across each subject helps to mitigate this concern.

In this context, it is disappointing that the work is motivated by situating it between two theoretical accounts of normalization (whether normalization affects perceptual representation, or not) that the manuscript claims its data to adjudicate. To quote, "These findings demonstrate that normalization is a highly robust phenomenon that affects representations at a level that precedes the mapping onto phonemes or higher level linguistic units."

The problem is that this is a theoretical straw man. In fact, the literature going back more than a decade makes clear that patterns of behavioral data just like those reported in the present manuscript can be elicited by nonspeech context precursors (which are not mapped to phonemes or phonetic features; Holt, 2005; Holt 2006; Laing et al. 2012; Huang & Holt, 2012). This is strong evidence for exactly the claim of the present paper — that normalization affects perceptual representation. (In this regard, it is curious that the manuscript does not take this more direct route of nonspeech contexts since it would eliminate the cumbersome and indirect linking hypotheses necessary in selecting electrodes to be 'encoding features' versus 'encoding categories.')

It is disappointing that the manuscript is written in a way that inflates the contributions of the work — presumably for impact, as it is hard to imagine that the excellent authors are unaware of the larger literature regarding this phenomenon.

The second problem is that when you acknowledge that there is an existing literature that demonstrates the manuscript's main claim (a contrastive normalization effect that affects pre-categorical perceptual representations), the primary advance of the present work is its observation of a contrastive effect of context in bilateral human temporal cortex. This is very much of interest, of course, and it nicely complements existing results from animal neurophysiology and human behavior. But, from an auditory neuroscience perspective it does not radically advance a new mechanistic model of normalization.

While it is really encouraging to see ECoG work advance to examining more dynamic aspects of speech processing, the primary claim is consistent with what has been observed in behavioral data (for 20 years as a general case, and nearly 15 using methods nearly identical to the behavioral task in the present study). These prior studies would directly predict the present results (and indeed, these predictions --including localization to cortex — have been described in prior publications not cited in the manuscript).

As a side note, I find it so odd that the literature review neglects to cite the literature I have mentioned above — these papers reach largely the same theoretical conclusions as the present manuscript and even predict the effect demonstrated here (cortical adaptation effects). E.g.,

Lotto & Kluender, 1998; <https://www.ncbi.nlm.nih.gov/pubmed/9628993> (first provided evidence for a pre-categorical contrastive mechanism in normalizing speech input according to context)

Holt, 2005; <https://doi.org/10.1111/j.0956-7976.2005.01532.x> (provides evidence of that targets are

encoded according to the 'mean' of the preceding context, as claimed — but not directly tested — in the present study)

Holt, 2006; <https://www.ncbi.nlm.nih.gov/pmc/articles/PMC1635014/> (provides the time course data that the manuscript alludes to, but does not test directly)

Laing et al 2012; <https://www.ncbi.nlm.nih.gov/pubmed/22737140> (directly relevant to the present study in demonstrating parallel effects of speech and non speech precursors in the same behavioral task).

Reviewer #2:

Remarks to the Author:

This is doubtlessly a well-crafted study: human electrocorticographic recordings are utilised to study how adaptation to a talker (aka speaker normalization) is reflected in perisylvian cortical responses to speech (ie vowels).

A (typically small) group of N=5 patients is studied here, with some of the not untypical problems arising, for example how to pool/analyse meaningfully the data from different subjects and a lot of electrodes.

I have trouble following the paper in its claims not in a fundamental way, but in the way the results are being presented as new or particularly important. I am well aware that this lack of enthusiasm is itself a contentious statement to make, but let me make more clear where I see a problem with this paper:

First, the authors start from the preposition that we know next to nothing how auditory cortex (AC) encodes speech sound or which level of abstraction is represented there. Not least from the work by the lead author Dr Chang himself, we actually have a quite rich understanding that AC is in fact not the "early" stage once thought. There are numerous studies from different species showing how perception leaves its imprint on AC activation (e.g. Perceptual Streaming, Micheyl&Rauschecker Neuron 2005; Lakatos et al., Neuron 2013; Attention more generally, Fritz/Shamma Nat Neurosci 2003 or Mesgarani & Chang Nature 2012; Vowels vs Speakers Formisano et al, Science 2008, to name but a few).

Quoting from Leonard and lead author Chang in TICS 2014, where they write about AC, even A1 proper, that "perhaps most important for understanding the early cortical stages of speech perception, is the fact that [A1 and surrounds] do not show strictly linear responses that can be characterized as faithful representations of the physical stimulus (a form of abstraction)."

I might thus be forgiven for taking the present submission's premise that (line 49 onwards) "A critical question that arises, then, is whether the feature-based representations in auditory cortex are normalized, or whether they continue to closely reflect the veridical acoustic properties of the input." as a bit of a strawman argument.

This discontent notwithstanding, the manuscript contains very fine strands of analysis, and the "neurometric" curves resembling the perceptual context effects of talker on vowel/first-formant percepts are amongst them.

However, also in analysis or at least in reporting these analyses, the paper largely falls short of the standards I would have expected fulfilled upon reading author list and abstract. Most conclusions seem to rest on p-values reported without any test statistic and measure of effect size. Where an occasional r value got reported, I was lost where the extraordinarily small p-values would result from (e.g., "r = -0.65; p = 1.3*10⁻⁶" in line 158f.). I can only guess that all electrodes were most likely pooled across subjects without further regard for the hierarchical structure of the data, yielding a data set that in all

likelihood violates the independence assumption for any linear-model usage but contains a lot of observations (subjects x electrodes, or even subjects x electrodes x stimulus levels).

While I might be wrong with my hunch here, it is obvious that the reporting leaves too much guess work to the reader. By all means, a much more thorough reporting of tests utilised and effect sizes observed would be necessary.

In sum, I read an interesting and in places elegant study on speaker normalization in human auditory cortex, but neither do I think the (overall quite clear-cut) results are surprising given what we know about auditory cortex and the way the entire central auditory pathway starting already in the brain stem is capable of stimulus-specific adaptation, nor was I able to fully follow the arguments from a technical–statistical vantage point.

Reviewer #3:

Remarks to the Author:

This is another of a series of interesting and revealing reports from the Chang Lab on the encoding of various aspects of speech in the human auditory cortex. I found all specific technical aspects of the work and data analyses excellent, and the text clear and concise. However, my main concern regarding the message of the MS is its interpretation of the results as a “speaker normalization” task when in fact the evidence for such an interpretation is lacking. Let me explain.

The design of the experiments inserts a context of a varying F1 depending on a “speaker”, and then asks about the perception of the subsequent vowel whose F1 was either above or below the context F1. Predictably the perception of the vowel varied depending on the context. As the authors make amply clear, this contrastive effect has been reported psychoacoustically before in many experiments, ranging from speech (Ladefoged), to simple stimuli such as tones and noise contexts. So the so-called contrast enhancement is a phenomenon that is well-known, well-studied, and has even be recorded physiologically in the brainstem, and of course in A1.

What the experiments here did is to repeat this phenomenon using a a high or a low F1 context (which can be described as a property of tall or short speakers), and then see the contrast enhancement. So this is really a study of contrast enhancement again. Just because one can describe the context F1’s as what would come out of different speakers does not really make it a speaker normalization task. To demonstrate speaker normalization, one needs to see its effects on a large variety of target vowels with a variety of F1 and F2’s and not just one that is precisely situated in an ambiguous zone. In short, while I totally find the results convincing and illustrative of contrast enhancement, I am not sure that it explains speaker normalization UNLESS that is, all speaker normalizations are based on contrast enhancement of subsequent perceptually ambiguous sounds. I really doubt that it is so simple.

Let me propose the following specific scenario to explain my concern. It is quite possible that a speaker’s voice creates a long term adaptive imprint at its average F1 and F2. This in turn would shift all subsequent vowels in all manner of ways up and down depending on where their F1’s and F2’s are. It is not obvious to me how such arbitrary effects are a sensible “normalization”! After all, what we want is a stable representation of the targets regardless of the context. Whatever the explanation is, it is not addressed in this MS. What is addressed is simply the contrastive effect one would see with vowels, tones, noise or just about any other context followed by an appropriately placed Stimulus.

In this light, I recommend that the authors tone down significantly their claims on speaker normalization, and perhaps instead emphasize the contrast enhancement, a phenomenon that may

not have been measured before specifically in the human auditory cortex, or perhaps not measured with vowels and speech-like stimuli in the human auditory cortex! They can make of course if they wish passing a reference to the fact that this phenomenon could be useful in the normalization of speakers.

Reviewer #1:

The major finding of the paper is that acoustically-identical speech sounds are encoded in a context-dependent manner in bilateral human temporal cortex. I am enthusiastic about the work and I find the study and its analysis to be well-conceptualized and carefully conducted. The data are sparse (few electrodes, few subjects), but the replication of the results across each subject helps to mitigate this concern.

In this context, it is disappointing that the work is motivated by situating it between two theoretical accounts of normalization (whether normalization affects perceptual representation, or not) that the manuscript claims its data to adjudicate. To quote, "These findings demonstrate that normalization is a highly robust phenomenon that affects representations at a level that precedes the mapping onto phonemes or higher level linguistic units."

The problem is that this is a theoretical straw man. In fact, the literature going back more than a decade makes clear that patterns of behavioral data just like those reported in the present manuscript can be elicited by nonspeech context precursors (which are not mapped to phonemes or phonetic features; Holt, 2005; Holt 2006; Laing et al. 2012; Huang & Holt, 2012). This is strong evidence for exactly the claim of the present paper — that normalization affects perceptual representation.

We thank the Reviewer for highlighting this way in which our previous manuscript could be improved to do justice to past literature on contrast enhancement in the auditory system. Related observations were also noted by Reviewers 2 and 3, and we took this point very seriously in our major revision of the manuscript. In particular, we have now substantially rewritten major sections of the manuscript, especially in the Abstract (**p. 2**) and Introduction (**pp. 3-5**; see also Results, **p. 7**, and Discussion, **p. 15-17**), to re-frame our argument in light of the significant past work on auditory contrast enhancement in human behavior.

We also thank Reviewer 1 for the suggested references, all of which are now included (among several others) in the revised manuscript. We believe this more complete treatment of contrast enhancement as a mechanism for normalization has dramatically improved the manuscript by reframing its central theoretical contribution – the first direct evidence of contrast enhancement of cortical vowel representations in human auditory cortex.

We strongly agree that the suggested reframing of our study better situates our work within the broader literature on auditory context effects in speech perception. At the same time, we also believe it is fair to point out in the manuscript that there is still some debate about the role of auditory contrast enhancement in speech sound normalization. For instance, we note evidence in the Discussion (**p. 18**) demonstrating that related normalization effects have also been observed in the absence of contrast enhancement, such as when listeners are merely *told* (e.g., during task instructions) that they are listening to a man or a woman (see, e.g., Johnson et al., 1999). We also refer to an influential literature that questions the relevance of auditory contrast effects for cognitive models of speech perception, both implicitly (Johnson, 2005; Goldinger 1998, **p. 7**), and also explicitly (**p. 16**: Viswanathan et al., 2010, 2013). We believe that framing our study in the context of contrast enhancement models in the Introduction, but later also addressing these other views, represents a balanced description of the existing literature, and in fact helps the reader to further recognize the study's novelty and significance.

... (In this regard, it is curious that the manuscript does not take this more direct route of nonspeech contexts since it would eliminate the cumbersome and indirect linking hypotheses necessary in selecting electrodes to be 'encoding features' versus 'encoding categories'.)

We agree with the reviewer that an analogous experiment, testing whether similar influences can be observed with nonspeech stimuli, would very likely provide another important contribution. In fact, we seriously considered taking this approach when designing the present experiment, but decided to focus on speech-sound-based effects because of the data limitations (e.g., feasible number of trials per subject) inherent to ECoG research. Specifically, previous work on vowel perception (by us and others) typically shows that behavioral normalization effects with *speech* contexts are larger and more reliable (replication at the participant level) than those with *nonspeech* contexts (e.g., Sjerps et al., 2011, 2012; Watkins & Makin 1996). Given these considerations, we opted to use speech contexts since we expected them to be most likely to result in robust context effects. If we had used nonspeech contexts and contrast enhancement had not been observed, it would have been unclear whether this was due to the use of a weaker

experimental manipulation, low power, or the lack of contrast enhancement in human auditory cortical representations.

However, to address the Reviewer's comment, we now mention (**p. 16**) that assessing neural contrast effects with nonspeech stimuli could be an important next step toward understanding contrast enhancement as a neural mechanism for normalization.

... It is disappointing that the manuscript is written in a way that inflates the contributions of the work — presumably for impact, as it is hard to imagine that the excellent authors are unaware of the larger literature regarding this phenomenon.

We hope that the substantial revisions to the manuscript will assuage the Reviewer's concerns by clarifying the novelty of our neurophysiological work and better situating it within the broader behavioral and theoretical literature (e.g., **p. 2**, **p. 3-5**, **p. 15-17**). We believe that the impact of this work stands on its own, and it is not nor was it ever our intention to inflate its contribution.

... The second problem is that when you acknowledge that there is an existing literature that demonstrates the manuscript's main claim (a contrastive normalization effect that affects pre-categorical perceptual representations), the primary advance of the present work is its observation of a contrastive effect of context in bilateral human temporal cortex. This is very much of interest, of course, and it nicely complements existing results from animal neurophysiology and human behavior. But, from an auditory neuroscience perspective it does not radically advance a new mechanistic model of normalization.

While it is really encouraging to see ECoG work advance to examining more dynamic aspects of speech processing, the primary claim is consistent with what has been observed in behavioral data (for 20 years as a general case, and nearly 15 using methods nearly identical to the behavioral task in the present study). These prior studies would directly predict the present results (and indeed, these predictions --including localization to cortex — have been described in prior publications not cited in the manuscript).

Substantial revisions to the Introduction (**pp. 3-5**) now address the existing literature on contrast effects in greater detail and earlier (in the Introduction), and we have also cited the publications suggested by the Reviewer. Like the Reviewer, we, too, thought that there was very good reason to predict that contrast enhancement might give rise to normalization effects in human auditory cortical speech representations. However, such effects have never previously been reported, and the issue was certainly not settled (as described above; see, e.g., Johnson, 2005; Goldinger 1998; Viswanathan et al., 2010, 2013). This prediction and major gap in empirical evidence is what led us to design and conduct the present experiment in the first place. We believe that demonstrating the role of contrast enhancement in normalization in human speech cortex processing represents an important contribution that, while it is consistent with the Reviewer's predictions, will surprise many other readers. These results significantly constrain the existing theoretical landscape for neural and cognitive models of speech processing.

As the Reviewer points out, this work also fills a critical gap in our understanding of normalization effects in speech perception as it presents the first report that directly connects two major existing bodies of work: one demonstrating contrast enhancement in animal neurophysiology, and another, based primarily on behavioral studies, suggesting that such contrastive mechanisms might operate in speech perception, too. We hope that the revised manuscript highlights this theoretical contribution more clearly, in addition to the empirical contributions described above.

... As a side note, I find it so odd that the literature review neglects to cite the literature I have mentioned above — these papers reach largely the same theoretical conclusions as the present manuscript and even predict the effect demonstrated here (cortical adaptation effects). E.g.,

Lotto & Kluender, 1998; <https://www.ncbi.nlm.nih.gov/pubmed/9628993> (first provided evidence for a pre-categorical contrastive mechanism in normalizing speech input according to context)

Holt, 2005; <https://doi.org/10.1111/j.0956-7976.2005.01532.x> (provides evidence of that targets are encoded according to the 'mean' of the preceding context, as claimed — but not directly tested — in the present study)

Holt, 2006; <https://www.ncbi.nlm.nih.gov/pmc/articles/PMC1635014/> (provides the time course data that the manuscript alludes to, but does not test directly)

Laing et al 2012; <https://www.ncbi.nlm.nih.gov/pubmed/22737140> (directly relevant to the present study in demonstrating parallel effects of speech and non speech precursors in the same behavioral task).

We apologize for these exclusions and thank the Reviewer for suggesting them. We agree with the Reviewer that our original manuscript neglected to cite some highly relevant work, including these studies, which was an oversight. We have added these and other crucial references to our previous manuscript's list of citations to work on normalization of vowels or vowel-like sounds, which more closely resemble our own study's vowel stimuli (e.g., Nearey, 1989; Kluender et al., 2003; Watkins, 1991; Stilp et al., 2010; Stilp et al., 2015; Sjerps et al., 2017).

Reviewer #2:

This is doubtlessly a well-crafted study: human electrocorticographic recordings are utilised to study how adaptation to a talker (aka speaker normalization) is reflected in perisylvian cortical responses to speech (ie vowels).

A (typically small) group of N=5 patients is studied here, with some of the not untypical problems arising, for example how to pool/analyse meaningfully the data from different subjects and a lot of electrodes.

I have trouble following the paper in its claims not in a fundamental way, but in the way the results are being presented as new or particularly important. I am well aware that this lack of enthusiasm is itself a contentious statement to make, but let me make more clear where I see a problem with this paper:

First, the authors start from the preposition that we know next to nothing how auditory cortex (AC) encodes speech sound or which level of abstraction is represented there. Not least from the work by the lead author Dr Chang himself, we actually have a quite rich understanding that AC is in fact not the "early" stage once thought. There are numerous studies from different species showing how perception leaves its imprint on AC activation (e.g. Perceptual Streaming, Micheyl&Rauschecker Neuron 2005; Lakatos et al., Neuron 2013; Attention more generally, Fritz/Shamma Nat Neurosci 2003 or Mesgarani & Chang Nature 2012; Vowels vs Speakers Formisano et al, Science 2008, to name but a few).

Quoting from Leonard and lead author Chang in TICS 2014, where they write about AC, even A1 proper, that "perhaps most important for understanding the early cortical stages of speech perception, is the fact that [A1 and surrounds] do not show strictly linear responses that can be characterized as faithful representations of the physical stimulus (a form of abstraction)."

I might thus be forgiven for taking the present submission's premise that (line 49 onwards) "A critical question that arises, then, is whether the feature-based representations in auditory cortex are normalized, or whether they continue to closely reflect the veridical acoustic properties of the input." as a bit of a strawman argument.

We agree with the Reviewer that the literature review and theoretical positioning of the previous manuscript did not provide enough detail. The result was an Introduction that came across as an overstatement of the work's novelty. As the Reviewer points out, it is already quite clear that A1, let alone parabelt auditory cortex, do not display a linear relation to the acoustic properties of the input. We recognize how the phrasing of our main question (testing whether parabelt auditory cortical responses "reflect the veridical acoustic properties of the input") was misguided in this way. We thank the Reviewer for pointing this out.

In response, in addition to the substantial rewrites and theoretical reframing described in response to Reviewer 1, we have also removed the specific phrasing that the reviewer points out (i.e., "veridical acoustic properties of the input"; **p. 2**). We also now explicitly acknowledge the key point the Reviewer is making in the Introduction: "...the STG's encoding of speech is not a strictly linear (veridical) encoding the acoustics; rather, it reflects some properties of abstraction, including categorical perception, relative encoding of pitch, and attentional enhancement" (**pp. 3-4**). The revised manuscript now focuses on the question of whether, in addition to reflecting these other properties of abstraction, representations are also contextually-normalized, and –specifically – whether they are affected by context in a contrastive direction, as predicted by models of contrast enhancement.

... This discontent notwithstanding, the manuscript contains very fine strands of analysis, and the "neurometric" curves resembling the perceptual context effects of talker on vowel/first-formant percepts are amongst them.

However, also in analysis or at least in reporting these analyses, the paper largely falls short of the standards I would have expected fulfilled upon reading author list and abstract. Most conclusions seem to rest on p-values reported without any test statistic and measure of effect size.

In the previous manuscript, only p-values were reported in the main text, with test statistics, parameters estimates, and effect sizes described in the Supplementary Materials. We agree that this did not provide enough immediate, comprehensive insight into the reliability and strength of the reported effects, so we now report this information in the main text along with the p-values (**pp. 6-9; pp. 12-13**).

*... Where an occasional r value got reported, I was lost where the extraordinarily small p-values would result from (e.g., "r = -0.65; p = 1.3*10⁻⁶" in line 158f.). I can only guess that all electrodes were most likely pooled across subjects without further regard for the hierarchical structure of the data, yielding a data set that in all likelihood violates the independence assumption for any linear-model usage but contains a lot of observations (subjects x electrodes, or even subjects x electrodes x stimulus levels).*

While I might be wrong with my hunch here, it is obvious that the reporting leaves too much guess work to the reader. By all means, a much more thorough reporting of tests utilised and effect sizes observed would be necessary.

As described in our previous comment, we have now provided more detailed descriptions of the statistical results in the main text. However, to respond to the specific observation of the Reviewer (the reported correlation across electrodes), it is true that, in the previous submission, we had computed correlations over electrodes pooled across participants (i.e., subjects * electrodes). As the reviewer rightly points out, this approach does not account for the hierarchical nature of the data (multiple electrodes are nested within individual subjects). In order to address this fair critique, we have updated the manuscript (**p. 9; p. 13**) to reflect our new approach, which accommodates the hierarchical nature of these observations (i.e., electrodes as observations within participants) by performing linear mixed-effects regression analyses. For example, technically speaking, the new regression formula for the analysis in Fig. 2d is: $t_{\text{CtxtEff}} \sim t_{\text{TargPref}} + (1 | \text{subj})$ (previously, it was: $t_{\text{CtxtEff}} \sim t_{\text{TargPref}}$). Because the coefficient of this regression is significantly negative, it means that, across subjects, electrodes display contrastive (negative) normalization. Thus, although the statistics are updated in the main text and Supplementary Materials, the results closely resemble our prior manuscript's reported analysis.

... In sum, I read an interesting and in places elegant study on speaker normalization in human auditory cortex, but neither do I think the (overall quite clear-cut) results are surprising given what we know about auditory cortex and the way the entire central auditory pathway starting already in the brain stem is capable of stimulus-specific adaptation, nor was I able to fully follow the arguments from a technical–statistical vantage point.

The Reviewer points out that the results presented here may not seem surprising given what we know about mechanisms such as SSA in brainstem processing. We agree that our findings align with those findings, but it is important to consider that the link between effects such as SSA or contrast enhancement (which have predominantly been based on animal physiology) and human speaker normalization has, to date, been based almost exclusively on behavioral research. The current study is, to our knowledge, the first to directly address the neurophysiological role of contrast enhancement in human speech sound normalization. We have substantially revised the manuscript to better clarify and highlight this novel contribution.

Reviewer #3:

This is another of a series of interesting and revealing reports from the Chang Lab on the encoding of various aspects of speech in the human auditory cortex. I found all specific technical aspects of the work and data analyses excellent, and the text clear and concise. However, my main concern regarding the message of the MS is its interpretation of the results as a "speaker normalization" task when in fact the evidence for such an interpretation is lacking. Let me explain.

The design of the experiments inserts a context of a varying F1 depending on a "speaker", and then asks about the perception of the subsequent vowel whose F1 was either above or below the context F1. Predictably the perception of the vowel varied depending on the context. As the authors make amply clear, this contrastive effect has been reported psychoacoustically before in many experiments, ranging from speech (Ladefoged), to simple stimuli such as tones and noise contexts. So the so-called contrast enhancement is a phenomenon

that is well-known, well-studied, and has even been recorded physiologically in the brainstem, and of course in A1.

What the experiments here did is to repeat this phenomenon using a high or a low F1 context (which can be described as a property of tall or short speakers), and then see the contrast enhancement. So this is really a study of contrast enhancement again. Just because one can describe the context F1's as what would come out of different speakers does not really make it a speaker normalization task. To demonstrate speaker normalization, one needs to see its effects on a large variety of target vowels with a variety of F1 and F2's and not just one that is precisely situated in an ambiguous zone. In short, while I totally find the results convincing and illustrative of contrast enhancement, I am not sure that it explains speaker normalization UNLESS that is, all speaker normalizations are based on contrast enhancement of subsequent perceptually ambiguous sounds. I really doubt that it is so simple.

Let me propose the following specific scenario to explain my concern. It is quite possible that a speaker's voice creates a long term adaptive imprint at its average F1 and F2. This in turn would shift all subsequent vowels in all manner of ways up and down depending on where their F1's and F2's are. It is not obvious to me how such arbitrary effects are a sensible "normalization"! After all, what we want is a stable representation of the targets regardless of the context. Whatever the explanation is, it is not addressed in this MS. What is addressed is simply the contrastive effect one would see with vowels, tones, noise or just about any other context followed by an appropriately placed Stimulus.

In this light, I recommend that the authors tone down significantly their claims on speaker normalization, and perhaps instead emphasize the contrast enhancement, a phenomenon that may not have been measured before specifically in the human auditory cortex, or perhaps not measured with vowels and speech-like stimuli in the human auditory cortex! They can make of course if they wish passing a reference to the fact that this phenomenon could be useful in the normalization of speakers.

The Reviewer elegantly points out that the relation between contrast enhancement and speaker normalization is far from direct, and that a complete understanding of the relation between them would involve testing a much larger and more diverse set of stimuli. Specifically, one would want to also include stimuli in which the F1 frequencies of the target do not fall exactly in an ambiguous region between the two context F1 trajectories. We agree with the Reviewer that the exact relation between contrast effects and normalization is incompletely addressed by the present study, but the main aim of the manuscript is to demonstrate, as a proof-of-principle, that contrast effects do arise in human auditory cortex, and these contrast effects are one key ingredient that could give rise, at least in part, to speaker-normalizing adjustments in speech sound representations. However, it is possible that there are configurations (ones not tested here) in which contrast enhancement may not result in normalization. It is clear that our experiment does not demonstrate that contrast enhancement is all there is to speaker normalization, and more research along these lines is necessary to demonstrate under what circumstances contrast enhancement may play a dominant role.

We hope that the major revisions to the manuscript emphasizing contrast enhancement have effectively responded to the Reviewer's concerns by situating our findings in the literature on contrast enhancement in the Abstract (**p. 2**) and Introduction (**pp. 4-5**; see also Results, **p. 7**, and Discussion, **p. 15-16**). Moreover, we point out in the manuscript that the F1 ranges of our target were 'ideally' situated with respect to the two contexts, and indicate that it is unclear whether the equally strong effects would have been found for other target stimulus ranges (**p. 18**). We then point out that the contrast enhancement effects that are demonstrated may only explain part of the normalization effects typically observed in speech perception (**p. 18**).

Reviewers' Comments:

Reviewer #1:

Remarks to the Author:

The authors have been responsive to each reviewer's comments and the manuscript is improved.

Reviewer #2:

Remarks to the Author:

The authors have provided a really insightful and careful revision. My view of this paper has thus changed considerably to the favourable. I would recommend it for publication, and alongside the rebuttal letter, would be happy for our exchange to appear in Nat Comms.

I find it particularly laudable, and effective, that results are now parsimoniously analysed in a linear-model framework. This will help set this a standard in the ECoG field.

A minor observation is that I was not sure whether the reported intercept betas and their statistics add anything; intercepts are hardly interpreted/useful by themselves (or am I missing something?). My suggestion is to add full tables for all/the most relevant calculated linear models to the supplements, which allows also for more liberty in reporting in-text only the conclusion-relevant (significant and non-significant!) effects.

Jonas Obleser

Reviewer #3:

Remarks to the Author:

I am fully satisfied with the revisions made. The authors have made sufficient changes to answer my concerns, and I am happy with the manuscript now as is.

Reviewer #1:

The authors have been responsive to each reviewer's comments and the manuscript is improved.

Reviewer #2:

The authors have provided a really insightful and careful revision. My view of this paper has thus changed considerably to the favourable. I would recommend it for publication, and alongside the rebuttal letter, would be happy for our exchange to appear in Nat Comms.

I find it particularly laudable, and effective, that results are now parsimoniously analysed in a linear-model framework. This will help set this a standard in the ECoG field.

A minor observation is that I was not sure whether the reported intercept betas and their statistics add anything; intercepts are hardly interpreted/useful by themselves (or am I missing something?). My suggestion is to add full tables for all/the most relevant calculated linear models to the supplements, which allows also for more liberty in reporting in-text only the conclusion-relevant (significant and non-significant!) effects.

We agree with the Reviewer that full statistical detail can be reported in a table. Unfortunately, following the manuscript checklist, we have reached the maximum number of 10 display items (2 tables, 8 figures). Hence, we have chosen to report only critical statistical detail in the main text, but to report full statistical detail of the tests in the supplementary materials.

Reviewer #3:

I am fully satisfied with the revisions made. The authors have made sufficient changes to answer my concerns, and I am happy with the manuscript now as is.